# Novel Ultrasound-Guided Hydrodissection with 5% Dextrose for the Treatment of Occipital Neuralgia Targeting the Greater Occipital Nerve

**DOI:** 10.3390/diagnostics14131380

**Published:** 2024-06-28

**Authors:** King Hei Stanley Lam, Daniel Chiung-Jui Su, Yung-Tsan Wu, Aeneas Janze, Kenneth Dean Reeves

**Affiliations:** 1The Board of Clinical Research, The Hong Kong Institute of Musculoskeletal Medicine, Hong Kong; 2Faculty of Medicine, The Chinese University of Hong Kong, Shatin, Hong Kong; 3Faculty of Medicine, The University of Hong Kong, Pokfulam, Hong Kong; 4Center for Regional Anesthesia and Pain Medicine, Wan Fang Hospital, Taipei Medical University, Taipei 110, Taiwan; 5Center for Regional Anesthesia and Pain Medicine, Chung Shan Medical University Hospital, Taichung 402, Taiwan; 6Department of Physical Medicine and Rehabilitation, Chi Mei Medical Center, Tainan 710, Taiwan; dr.daniel@gmail.com; 7Department of Physical Medicine and Rehabilitation, Tri-Service General Hospital, School of Medicine, National Defense Medical Center, Taipei 114, Taiwan; crwu98@gmail.com; 8Integrated Pain Management Center, Tri-Service General Hospital, School of Medicine, National Defense Medical Center, Taipei 114, Taiwan; 9Department of Research and Development, School of Medicine, National Defense Medical Center, Taipei 114, Taiwan; 10Walter Reed National Military Medical Center, Bethesda, MD 20889, USA; ajanze@yahoo.com; 11Private Practice PM&R and Pain Management, 4840 El Monte, Roeland Park, KS 66205, USA; deanreevesmd@gmail.com

**Keywords:** occipital neuralgia, suboccipital headache, greater occipital nerve, ultrasound-guided, hydrodissection, 5% dextrose in sterile water

## Abstract

Background: Occipital neuralgia is a debilitating condition, and traditional treatments often provide limited or temporary relief. Recently, ultrasound-guided hydrodissection of the greater occipital nerve (GON) has emerged as a promising minimally invasive approach. Objectives: To describe two novel ultrasound-guided hydrodissections with 5% dextrose for GON and discuss their advantages, disadvantages, and considerations. Methods: Two cases are reported. Case 1 describes a lateral decubitus approach for hydrodissecting the GON between the semispinalis capitis (SSC) and obliquus capitis inferior (OCI) muscles. Case 2 details a cranial-to-caudal approach for hydrodissecting the GON within the SSC and upper trapezius (UT) muscles when the GON passes through these two muscles. Results: Both patients experienced significant and sustained pain relief with improvements in function. Conclusions: Ultrasound-guided GON hydrodissection using 5% dextrose is a promising treatment for occipital neuralgia. The lateral decubitus and cranial-caudal approaches provide additional options to address patient-specific anatomical considerations and preferences.

## 1. Introduction

Occipital neuralgia is a common and debilitating condition characterized by pain and tenderness in the suboccipital region, with referral to the head along the distribution of the greater, lesser, or third occipital nerves [1]. This type of headache can be caused by a variety of factors, including muscle tension, joint inflammation, and nerve entrapment. Diagnosis of occipital neuralgia requires severe, episodic, and sharp shooting or stabbing pain in one or more occipital nerve distributions, episodes that last only minutes, and relief after local anesthetic block [2]. Allodynia or dysesthesia may be present in the nerve distribution [3]. Tinel’s sign is commonly found [4]. A recent study of 800 patients who presented with a chief complaint of headache found 25% met criteria for occipital neuralgia [5]. It is important to note, however, that up to 85% of patients with occipital neuralgia may have an additional headache type [5]. For that reason, accompanying tension, migraines, cluster headaches, other neuralgia, or mechanical neck pain causing referred pain to the head should be kept in mind for treatment consideration as well, to maximize the overall functional impact of treatment [6].

The traditional treatment of occipital neuralgia typically involves a combination of treatments, including pharmacological and nonpharmacological management, e.g., nonsteroidal anti-inflammatory drugs (NSAIDs), muscle relaxants, opioids, tricyclic antidepressants, postural exercise, manual therapy, or modalities such as TENS [7]. Perineural injections (blocks) using local anesthetics or corticosteroids have been extensively utilized. A recent meta-analysis of nerve block efficacy for occipital headache compared to baseline status, and compared to other treatments was performed by Evans et al. [8]. They reported significant improvement in frequency and severity of occipital headaches compared to baseline for 6 weeks and 6 months, respectively, with a mean improvement in pain severity of 40–45%. In active treatment comparison trials, cryoneurolysis, pulsed radiofrequency treatment, and GON treatment with botulinum toxin A (BTX) outperformed anesthetic blocks at 6 months for the magnitude of pain improvement [8]. Among ablation techniques, pulsed radiofrequency or cryoablation preserve nerve architecture better than thermal radiofrequency or chemical ablation, with less potential for hypesthesia or dysesthesia [6], but long-term follow-up studies are lacking [9]. BTX injection has a better side effect profile but requires repetition indefinitely [6]. Surgical approaches such as partial resection of the obliquus capitis or C2 gangliotomy have had success but carry the risk of neuromas and causalgia [6]. Two studies have been reported with favorable results and minor side effects, but the retrospective nature of these data collection studies, with inherent bias, selective recall, incomplete data capture, and a lack of formal side effect analysis, limits the strength of conclusions [10,11]. 

Occipital neuralgia almost always results from compression of the greater occipital nerve (GON), lesser occipital nerve, or third occipital nerve at one or more points along their course, and the GON is the primary source in 90% of cases [12,13]. The GON is the dorsal rami of the second cervical nerve root. After branching out from the C2 nerve root, it wraps around the obliquus capitis inferior (OCI) muscle, ascends between the OCI and semispinalis capitis (SSC) muscle, and then between the rectus capitis posterior major (RCPMa) muscle and SSC. Then it pierces through the SSC and upper trapezius (UT) to surface the subcutaneous tissues in the occipital area [14]. Consistent with the complicated course of the GON, technique appears to play a significant role in GON block outcomes. Using ultrasonographic or fluoroscopic guidance to block the GON in the suboccipital compartment enhances the accuracy of injection [15] and durability of the benefit [16,17] compared to the classical technique for GON block at the superior nuchal line. A guided suboccipital compartment approach has been recommended for future comparative studies between GON block and Botulinum toxin injection or neurolysis [8]. 

Recently, ultrasound-guided hydrodissection of the GON has emerged as a promising treatment option for occipital neuralgia, and three case reports have been published in recent years [18,19,20]. Hydrodissection is a minimally invasive procedure that involves injecting a small amount of fluid, typically 5% dextrose in sterile water (D5W) or normal saline (NS), with or without local anesthetic solution, into the tissue surrounding the GON to release any adhesions or scar tissue that may be compressing or irritating the nerve. All papers proposed performing ultrasound-guided hydrodissection with the patient lying prone. Rose et al. [18] and Kaga et al. [20] proposed hydrodissecting the GON between the SSC and OCI with a lateral to medial in-plane approach and an out-of-plane approach, respectively. Ryan et al. proposed hydrodissecting even deep to the OCI, targeting the C2 nerve root at the C2 transverse process [19], which we believe should not be the “transverse process” but rather the pedicle of C2.

We have observed some patients who cannot lie prone. Additionally, the GON entrapment may be inside the SSC and UT when the nerve pierces through these two muscles. Herein, we describe two different approaches in detail that are appropriate for general use and feasible for use with prone positioning difficulty or entrapment inside SSC and UT, elaborating on their advantages and disadvantages. D5W was utilized as the injectate in these two cases, rather than anesthetic injection. This is related to the apparent therapeutic effect of dextrose itself [21], beyond that of a mechanical effect of injection [22], which will be addressed in the Section 3.

## 2. Case Presentation

### 2.1. Case 1

A 45-year-old woman, presented with an 18-month history of insidious-onset occipital neuralgia. The pain was described as stabbing and lancinating, with occasional radiation from the right suboccipital region to the vertex. She reported that computer work and poor sleep exacerbated the pain. Her worst pain occurred after a prolonged workday, reaching 8/10 on the numerical rating scale (NRS). Head pain significantly affected her concentration span, e.g., reading news on her smartphone and looking at the computer content, disturbing her work. Oral medication such as prednisone, diazepam, amitriptyline, acetaminophen, or ibuprofen, and repeated acupuncture sessions and trigger point injections with corticosteroid and local anesthetics provided only partial and temporary relief, with pain recurrence to the same level. The patient underwent multiple sessions of TENS, manual therapy, and magnetic and infrared therapy administered by the physiotherapist, but these modalities failed to provide any relief. The neck disability index (NDI) [23] was 20/50, indicating moderate disability [23]. Physical examination revealed tender trigger points along the right lateral cervical spine and occipital base, particularly over the upper SSC. The neuromuscular examination was intact. The digit span test for short-term memory revealed a forward recall of 5 digits and a backward recall of 4 digits [24], below normal limits. Neuroimaging studies, including an MRI of the cervical spine and brain, did not show any abnormalities.

She provided informed consent to undergo an ultrasound-guided hydrodissection with D5W of the GON. Due to her severe allergic rhinitis and reported difficulty lying prone, a lateral decubitus approach was selected for her. The patient underwent ultrasound-guided hydrodissection of the GON inside the fascial plane between the SSC and OCI with 20 mL of D5W without local anesthesia [25] (Figure 1 and Video 1 [25]). The procedure was well-tolerated, and she did not experience any side effects. The patient’s pain dropped from 8/10 to 0/10 immediately after the hydrodissection, and her concentration span improved, as noted by a forward recall of 8 digits and a backward recall of 6 digits on the digit span test for short-term memory [24]. She received regular follow-up, initially weekly, then biweekly, and monthly through 6 months, with no additional treatment required. Her pain score sustained 0–1/10 at 6 months, and her NDI at 6 months was 4/50, consistent with no disability. She was asked to re-contact the clinic if pain recurred. Given a minimal clinically important difference (MCID), which is the smallest change in a treatment outcome that an individual patient would identify as important [26], for the NDI in those without radicular upper extremity pain of 5.5 points, this would be approximately 2.5 MCID levels of change, or an 80% improvement [27].

### 2.2. Case 2

A 50-year-old woman presented with a one-year history of suboccipital pain that had developed six months after a whiplash injury in a car accident. She described the pain as a baseline dull ache and tightness with frequent exacerbations to severe, episodic, and sharp shooting or stabbing pain in the right GON distributions, episodes that last only minutes, and relief temporarily after local anesthetic block. The pain would lead to brain fog and blurring of vision, which significantly affected her work. An ophthalmological consultation revealed no abnormalities in her eyes. The pain increased during each workday, reaching an NRS of 9/10. The NDI was 24/50, indicating moderate disability. Physical examination revealed trigger points along the right lateral cervical spine and occipital base, particularly in the upper SSC and UT. The neuromuscular examination was normal. MRI imaging showed mild degeneration of the lower cervical spine but no abnormalities in the upper cervical spine or brain.

Previous treatments, including amitriptyline, diazepam, sleep management, and acupuncture targeting the suboccipital muscles, provided temporary, partial relief, but the pain would recur after each workday. Two courses of TENS, manual therapy, and magnetic and infrared therapy by the physiotherapist did not provide any relief for the patient. An ultrasound-guided GON block by another pain physician between SSC and OCI, using 1 mL of 2% lidocaine and 40 mg of triamcinolone, provided temporary pain relief for a few hours, with the pain returning the following day after work. Together with the swelling and tenderness of the SCC and UT, where the GON passes through these muscles, and considering the limited success of the previous interventions, she provided informed consent to undergo an ultrasound-guided hydrodissection using 20 mL of D5W without local anesthetic of the long-axis of the GON within the SSC and UT as well as between the SSC and RCPMa [25] from cranial to caudal approach (Figure 2 and Video 2 [25]). She tolerated the procedure well, with only mild discomfort (1–2/10 NRS) during the injection. Upon completion of the procedure, the NRS dropped from 9/10 to 1/10, and her brain fog and blurring of vision were immediately resolved. She received regular follow-up, initially weekly, then biweekly, and monthly up to 6 months, with no further treatment required. The NRS sustained at 0–1/10, with an NDI of 3/50 at 6 months, and instructions to re-contact with pain recurrence. This improvement in the NDI represents 3.8 times the MCID, or 85%.

## 3. Discussion

### 3.1. Ultrasound-Guided Greater Occipital Nerve (GON) Hydrodissection with 5% Dextrose in Water in the Lateral Decubitus Position

Dr. Rose’s seminal article [18] proposed a lateral to medial approach, with the patient lying prone, for hydrodissecting the GON between the SSC and OCI muscles. However, we suggest considering a medial to lateral approach with the patient lying in lateral decubitus, as depicted in Figure 1 and Video 1, if the patient cannot tolerate lying prone [25]. This positioning will also provide better visualization and a more ergonomic approach to injection for the physicians.

Video 1: Ultrasound-guided hydrodissection of the greater occipital nerve (GON) between the semispinalis capitis (SSC) and obliquus capitis inferior (OCI) muscles, with a posteromedial to anterolateral approach in the lateral decubitus position [25]. The labeled sonoanatomy, as shown in Figure 1, has been embedded in the still image of the video.

https://www.dropbox.com/scl/fi/6i85zm2y2ouutv3l5d22r/GON-HD-between-SCC-and-OCI-lateral-decubitus.mp4?rlkey=9bsaie2l4vvd5vdbsurv5u2pl&dl=0.

#### 3.1.1. Technical Notes for the Hydrodissection of the GON with 5% Dextrose in Water in a Lateral Decubitus Position

Patient selection: Patients fulfill the diagnostic criteria as described in the introduction for occipital neuralgia, with other differential diagnoses excluded. They have failed other conservative treatments. The local tenderness is felt mainly over the OCI (usually slightly lateral than the GON entrapped within the SCC and UT groups), and compression may reproduce their occipital neuralgia-type pain between the SCC and OCI but not within the SCC and UT. There is no swelling or tenderness in the SCC or UT. Ultrasound shows swelling of the GON compared to the normal, non-painful side between the SCC and OCI, or a cross-sectional area exceeding the upper limit of 3 mm^2^ [28]. If they have swelling and tenderness of the SCC and UT (more medial than the entrapment between the SCC and OCI group), it means the GON may also be entrapped within the SCC and UT, indicating that the GON may also need to be hydrodissected using the cranial-caudal technique described below.Patient’s Position: The patient is typically positioned in a lateral decubitus position with the GON under treatment facing upward. This positioning allows for an alternative treatment option if the patient is unable to lie prone. From the patient’s perspective, it is generally more comfortable if the needle comes from behind, especially when they are fully awake, as it prevents them from seeing the needle.The Ultrasound Machine and Probe: The ultrasound machine is placed in front of the patient, facing the treating physician, to optimize ergonomics. The ultrasound used in this manuscript was the Logiq S7 (General Electric, Boston, MA, USA). A linear transducer with a broad frequency range (GE L3-12 D, General Electric, Boston, MA, USA) is used, which provides sufficient penetration and resolution. It is positioned on the transverse plane, aligned with the long axis of the OCI, and emits ultrasound waves to visualize the underlying anatomy.Identifications of Important Sonoanatomy: Obliquus Capitis Inferior Muscle: This is a striated muscle that originates from the spinous process of the axis (C2) and inserts into the posterior aspect of the transverse process of the atlas (C1) (Figure 1). Semispinalis Capitis Muscle: The SSC muscle is a deep neck muscle visualized in the short-axis view. It is located superficial to the OCI and underneath the splenius capitis and UT. The UT and sternocleidomastoid muscles are the most superficial muscles in that region (Figure 1). Greater Occipital Nerve: The GON is identified as a hypoechoic oval structure situated between the SSC and OCI (Figure 1). In cases where there is swelling or inflammation, the GON may appear enlarged compared to the normal side.Needle: A 25 G or 27 G 2-inch hypodermic needle is typically used. It is inserted under ultrasound guidance, approaching the GON for the hydrodissection procedure both above and below the GON (Figure 1 and Video 1 [25]).Inject: D5W without local anesthetic.Method: Appropriate disinfection preparation uses 2% chlorhexidine in 75% isopropyl alcohol. It is also advisable to use the same disinfectant solution as the contact media for the ultrasound transducer [29]. The needle entry point is anesthetized with 0.5% lidocaine to create a skin wheel. With ultrasound guidance, a 25 G or 27 G 2-inch hypodermic needle is inserted using an in-plane technique. The needle tip is advanced to the site of the GON using a hydrodissection technique. Injection of D5W during needle advancement pushes away any soft tissue in front of the needle tip and creates a halo, allowing the needle tip to follow from the subcutaneous fat to the site and the whole tract of the GON to be hydrodissected. The needle is guided to hydrodissect the GON between the SSC and OCI, and the soft tissue is completely hydrodissected above and below the GON, with the GON completely encircled with anechoic fluid. Typically, 20 to 30 mL of D5W without local anesthetic will be used for one side of GON hydrodissection, from skin entry to complete hydrodissection of the GON, resulting in the patient’s pain being relieved.

#### 3.1.2. Advantages of the Ultrasound-Guided Greater Occipital Nerve (GON) Hydrodissection Procedure in the Lateral Decubitus Position

Patient Comfort:-The lateral decubitus position is generally more comfortable for the patient compared to lying prone, especially if they have difficulty lying prone.-Patients may feel more at ease when the needle comes from behind, as they cannot directly see the needle approach.-Some patients reported feeling very nervous when lying prone and not knowing what to expect during the procedure.Improved Visualization:-With the patient in the lateral decubitus position and the ultrasound probe placed in the long axis of the OCI, the transducer can be optimally positioned parallel to the OCI muscle fibers to provide clear visualization of the target anatomy, including the OCI, GON, the relevant arteries, and the C2 nerve root.-The utilization of the in-plane technique, with the needle entering from a posterior to anterior direction, offers a distinct advantage in terms of visualization. This technique allows for a highly clear and detailed visualization of the needle when hydrodissecting above and below the GON, between the OCI and SSC. This is because the transducer aligns more parallelly with the needle, enhancing the precision and accuracy of the procedure.-This approach can provide a clearer needle trajectory compared to the traditional prone position, even when injecting the C2 nerve root over the pedicle.-Even with the traditional prone position, the needle entry point utilized in the lateral decubitus approach, at the C2 spinous process, may offer better needle visualization and trajectory. This is because the needle would be more parallel to the transducer, enhancing the precision of the procedure.Learning Curve:-The improved visualization of the anatomical structures and needle trajectory may provide clinicians with a shorter learning curve to properly visualize the target anatomy and guide the needle accurately.Procedural Safety:-The lateral decubitus position can provide better visualization of the suboccipital structures while also relaxing the patient, which may significantly reduce the risk of inadvertent vascular puncture or injection compared to the prone position.-The improved visualization of the target anatomy and needle trajectory can help the clinician accurately guide the needle to the desired location, minimizing the risk of nerve injury.

#### 3.1.3. Disadvantages of the Ultrasound-Guided GON Hydrodissection Procedure in the Lateral Decubitus Position

Patient Positioning:-While the lateral decubitus position may be more comfortable for some patients, it may still be difficult for patients with limited neck mobility or other physical limitations. Patients with pain over the contralateral shoulder and hip may not be able to tolerate the lateral decubitus position.

### 3.2. Cranial to Caudal Approach of Ultrasound-Guided Hydrodissection with 5% Dextrose of the Greater Occipital Nerve inside the Semispinalis Capitis and Upper Trapezius Muscle

Current literature does not provide any approach to hydrodissecting the GON inside the SSC and UT muscles. However, in our clinical experience, we have observed a higher prevalence of patients with GON entrapment within the tight SSC and UT muscles when the GON passes through these muscles before emerging in the subcutaneous tissues of the occipital region. Many patients exhibit swelling and tenderness over the more cranial aspects of the SSC and UT, which may indicate the presence of trigger points in these muscles. These trigger points are highly likely to entrap the GON and/or the third occipital nerve (TON) just before they surface in the subcutaneous tissues. As far as we know, this is the first manuscript that describes incorporating this alternative cranial-to-caudal approach for hydrodissecting the GON within the UT and SSC muscles to help address cases where the nerve is entrapped in this region, as shown in Figure 2 and Video 2.

Video 2: Cranial to Caudal Approach of Ultrasound-Guided Hydrodissection of the Greater Occipital Nerve within the Semispinalis Capitis and Upper Trapezius Muscles [25]. The labeled sonoanatomy, as shown in Figure 2 has been embedded in the still image of the video.

https://www.dropbox.com/scl/fi/vz9sxnd25tv8wk0y4pf2j/GON-HD-Cranial-to-Caudal-with-Labelling_UT-updated.mp4?rlkey=qaeeh931q90n8r49dcmz8yeab&dl=0.

#### 3.2.1. Technical Notes of the Cranial Approach of Ultrasound-Guided Hydrodissection with 5% Dextrose in Water of the Greater Occipital Nerve inside the Semispinalis Capitis and Upper Trapezius Muscle

Patient selection: Patients fulfill the diagnostic criteria as described in the introduction for occipital neuralgia, with other differential diagnoses excluded. They have failed other conservative treatments. Local tenderness and swelling are felt mainly over the UT and SCC (usually more medial than the GON entrapped between the SCC and OCI groups). Compression of the SCC and UT may reproduce their occipital neuralgia-type pain, but not compression between the SCC and OCI, and there is no swelling and tenderness between the SCC and OCI. Ultrasound may show swelling of the GON as a hypoechoic structure passing through the SCC and UT, compared to the normal, non-painful side, or a cross-sectional area exceeding the upper limit of 3 mm^2^. However, the cross-sectional area of the GON within the SCC and UT is very difficult to measure accurately due to the similarity of echogenicity with the surrounding muscles. If they have swelling and tenderness of the SCC and OCI (more lateral than the entrapment within the SCC and UT group), it suggests an additional entrapment site of the GON between the SCC and OCI, indicating a need to add hydrodissection using the posteromedial to anterolateral approach technique described above (in Section 3.1).Patient’s Position: The patient is typically positioned in a prone position with the head and neck fully flexed to allow for optimal ultrasound scanning and guided injection. This position also helps to minimize the curvature in the suboccipital region, which can otherwise make the procedure challenging to perform.The Ultrasound Machine and Probe: The ultrasound machine is placed by the side of the treatment table with the monitor in front of the treating physician for optimal ergonomics. The ultrasound used in this manuscript was the Logiq S7 (General Electric, Boston, MA, USA). A linear transducer with a broad frequency range (GE L3–12 D, General Electric, Boston, MA, USA) was used, which provided sufficient penetration and resolution. The transducer was positioned in the sagittal plane, aligned with the long axis of the SSC and UT muscles.Identifications of Important Sonoanatomy: Upper Trapezius Muscle: The UT muscle is located superficially in this region, beneath the adipose tissue and fascia. It is a thin layer of striated muscle. Semispinalis Capitis Muscle: The SSC muscle is a deep neck muscle that is visualized in a long-axis view. It is located superficially to the rectus capitis superior muscle and underneath the UT and splenius capitis muscles. Greater Occipital Nerve: The GON is identified as a hypoechoic fascicular structure that passes through the SSC and UT muscles. In cases where there is swelling or inflammation, the GON may appear enlarged compared to the normal side.Needle: A 25 G or 27 G 2-inch hypodermic needle is inserted under ultrasound guidance to access the GON for hydrodissection above and below the nerve.Inject: D5W without local anesthetic.Method: Due to the presence of hair in this region, a larger amount of 2% chlorhexidine in isopropyl alcohol is advised to be used as the disinfection agent. The disinfectant solution may need to be applied several times to ensure proper disinfection. It is also advised to use the same disinfectant solution as the contact media for the ultrasound transducer. The needle entry point is anesthetized with 0.5% lidocaine to create a skin wheel. With ultrasound guidance, a 25 G or 27 G 2-inch hypodermic needle is inserted. The physician keeps injecting with D5W with needle advancement to push away any soft tissue in front of the needle tip and create a halo, allowing the needle tip to follow from the subcutaneous fat to the site. This allows for the whole tract of the GON to be hydrodissected, within the UT and SCC, and between the SCC and RCPMa, as the GON passes through these anatomical structures. The soft tissue is completely hydrodissected above and below, lateral, and medial to the GON, with the GON completely encircled with anechoic fluid. During the procedure, in-plane and out-of-plane techniques will be used interchangeably to validate the needle position. Typically, 20 to 30 mL of D5W without local anesthetic will be used for one side of GON hydrodissection, from skin entry to complete hydrodissection of the GON, resulting in the patient’s pain being relieved.

#### 3.2.2. Advantages of the Cranial to Caudal Approach for Ultrasound-Guided Hydrodissection of the Greater Occipital Nerve (GON)

True Access to Nerve Entrapment Sites: This approach is the first ultrasound-guided hydrodissection approach to truly target the GON as it passes through the SSC and UT muscles, which are other common sites of nerve entrapment [25]. Currently, no other proposed non-surgical treatments are effective [30].Potential for More Comprehensive Treatment: By addressing the GON within the SSC and UT muscles and between the SCC and RCPMa, this technique can complement the previously described lateral decubitus approach to hydrodissect the GON between the SSC and OCI muscles, as described in Section 3.1. This combination of techniques may offer a more comprehensive management strategy and treat a longer length of the affected portion of the GON in patients with occipital neuralgia.Improved Visualization: The prone position with neck flexion helps minimize the curvature of the suboccipital region, allowing for better visualization of the underlying anatomy using the high-frequency linear ultrasound transducer.

#### 3.2.3. Disadvantages of the Cranial Approach for Ultrasound-Guided Hydrodissection of the Greater Occipital Nerve (GON)

Patient Positioning Challenges: The prone position with neck flexion may be less comfortable for some patients, particularly those with pre-existing neck or back problems. Maintaining this position throughout the procedure can be challenging.Increased Technical Difficulty: The curved anatomy of the suboccipital region and the depth of the GON within the SSC and UT muscles may make the ultrasound-guided needle placement more technically demanding and increase the potential risks, e.g., nerve injury, compared to the posteromedial-to-anterolateral approach described in Section 3.1.

### 3.3. Selection of Injectate for Ultrasound-Guided Hydrodissection of the Greater Occipital Nerve (GON)

The selection of injectate for ultrasound-guided hydrodissection of GON varies in the literature. Rose et al. [18] reported using 1 mL of 1% lidocaine plus 9 mL of NS; Ryan et al. [19] used 9 mL of NS and 1 mL of 40 mg/mL triamcinolone; and Kaga et al. [20] used only 5 mL of 0.75% ropivacaine. To the best of our understanding, Lam [25] is the first to report the use of D5W as the injectate for hydrodissecting GON. In recent years, many experienced practitioners worldwide have found that D5W demonstrates greater efficacy for hydrodissection compared to NS and corticosteroids [31,32,33,34,35,36,37].

The GON is a sensory nerve that sends nociceptive signals, among other sensory modalities, to the dorsal horn of the spinal cord. When the nerve itself and/or the nerve bed are sensitized due to chemical (e.g., inflammatory substances) or mechanical (scar tissue) stimuli, it may cause pain and tenderness in the suboccipital region. The mechanism by which ultrasound-guided hydrodissection of the GON with D5W relieves suboccipital headache is thought to be related to the release of compression or irritation of the nerve [31,32,36], as well as a direct therapeutic effect of D5W on nerves [33,34,35].

### 3.4. Mechanism of Action of Glucose

The evidence for a direct ameliorative effect of dextrose on neurogenic pain is accumulating in three primary areas. One proposed neurogenic mechanism is the “glycopenic hypothesis”. Kim et al. found dextrose superior to saline or 0.5% lidocaine for trigger injection and proposed an “energy benefit” from the use of dextrose at 12.5% in the injectate [38]. MacIver et al. demonstrated that C fibers exposed to the removal of D-glucose (dextrose) fire 650% faster within 20 min and return to a normal firing rate with the reintroduction of dextrose [39]. Dr. John Lyftogt proposed that high-energy-requiring C fibers use up their available energy stores through enhanced firing rates in the presence of chronic pain, resulting in a perineural glycopenic state [40]. As glycolysis is the primary source of ATP production [41], nerves may have insufficient ATP to power their sodium-potassium pump, which is critically required for repolarization. Sensory nerves that do not reach a minus 70 millivolt transmembrane potential will fire much more easily and can fire constantly, perpetuating chronic pain [42]. Although dextrose upon injection only donates 2 ATP initially via the first glycolytic step, that occurs in seconds and may explain the immediate benefit of dextrose injection.

Multiple observations have suggested that dextrose increases the level of substance P (SP). Although substance P (SP) has been characterized as promoting nociception [43], that effect appears to be confined to the spinal cord [44]. SP reduced hyperalgesia and allodynia via supraspinal injection in a rat inflammatory pain model [45], reduced nociception in an acid-induced pain model [46], and improved neuropathic pain after intravenous administration in a neuropathic pain model [47]. SP activates opioid receptors via NK1 receptor binding [48] and is being considered a candidate for neuropathic pain treatment in human trials [47]. Han et al. showed that dextrose likely exerts its analgesic effect by inducing the release of substance P through the activation of the acid-sensing ion channel 1a (ASIC1a) [49]. Topol et al. demonstrated analgesia after injection of dextrose in painful osteoarthritic knees, reducing pain with walking by 4.2 on the 0–10 NRS within 20 min, accompanied by a rise in synovial fluid SP by 112% at one week [50].

Other studies have demonstrated an anti-inflammatory effect of dextrose on nerves. An in vitro study by Wu et al. showed that dextrose decreased the inflammatory markers IL-6 and IL1B in TNF-a-exposed nerve cells in a dose-dependent manner [34]. Similarly, Cherng et al. found that TNF-a-exposed nerve cells exhibited increased survival and decreased reactive oxygen species (ROS) in response to increasing concentrations of dextrose [32]. Topol et al. showed that medium-term (3-month) levels of the important degenerative neuropeptide Y (NPY) dropped markedly in synovial fluid in human stage IV osteoarthritic knees [50] after monthly dextrose injection.

Consistent with potential dextrose effects on C fiber energetics, SP levels, inflammatory markers, or other mechanisms as yet unexplored, we propose that dextrose has a multifaceted mechanism by which it alleviates pain and reduces inflammation. The role of dextrose in modulating key neuropeptides and ion channels presents exciting avenues for further research and clinical applications.

### 3.5. Potential Limitations of Ultrasound-Guided Greater Occipital Nerve (GON) Hydrodissection

Technical Challenges:-Anatomical variation in the course and branching pattern of the GON can make it difficult to precisely locate and target the nerve during the hydrodissection procedure.-Identification of the exact fascial planes and muscle layers surrounding the GON may be challenging, especially in patients with previous injuries and surgery involving the musculature or adipose tissue in the suboccipital region.-While hydrodissection of the GON shows promising results, long-term studies with larger sample sizes are needed to confirm its efficacy and safety.-The success of the procedure relies heavily on accurate needle placement, which requires expertise in ultrasound-guided injections.-The presence of excessive hair in the suboccipital region and the curvature of the suboccipital region may make the ultrasound-guided procedure in this area challenging, especially with the use of prone positioning.Patient Factors:-Patients with limited neck mobility or difficulty lying in the required position (e.g., intolerance to the prone position) may have difficulty tolerating the procedure.-Patients with significant cervical spine pathology, such as severe degenerative changes or instability, may not be suitable candidates for the hydrodissection procedure.-It is important to consider the potential underlying causes of occipital neuralgia when assessing the effectiveness of hydrodissection for long-term pain relief. In cases where the neuralgia is secondary to conditions such as arthritis or spinal stenosis, hydrodissection may not provide sustained relief. Additionally, if the impingement of other occipital nerves, such as the lesser and/or third occipital nerves, is contributing to the neuralgia, targeting only the GON through hydrodissection may not be sufficient for effective pain management.-Empirical observations suggest that the response to hydrodissection can vary significantly between individuals, and large-scale studies are needed to standardize this technique, its protocol, and indications for this procedure.Incomplete or Transient Pain Relief:-While the procedure has shown promising results, in some cases, the pain relief may be incomplete or temporary, and require repeat hydrodissection or other interventions.-The long-term durability of pain relief with GON hydrodissection is still being evaluated, and the optimal frequency of repeat procedures is not yet well-established.Possible Side Effects of Ultrasound-Guided GON Hydrodissection Using D5W Without Local Anesthetic:-The potential side effects of the ultrasound-guided GON hydrodissection described in this manuscript, according to the authors’ experiences, are primarily related to the needling procedure itself. Systemic effects or side effects from the unintentional flow of fluid to surrounding structures are not expected using D5W as the injectate rather than a local anesthetic.-Its side effects profile is similar to, but considerably less severe than, the ultrasound-guided GON block. This is because a hydrodissecting technique is used, which theoretically pushes the nerves and vascular structures away from the needle tip, further reducing the chance of inadvertent vascular puncture and nerve irritation.-Temporary discomfort may be possible due to the pressure effect of the solution.

## 4. Conclusions

These case reports suggest that GON hydrodissection may be a clinically useful approach to occipital neuralgia treatment. Careful patient selection and consideration of the underlying pathology causing the occipital neuralgia are crucial to optimizing the outcomes of this technique. Patients should be thoroughly evaluated for potential anatomical variations, other contributing pain generators, and their ability to tolerate the procedure.

While retrospective results are promising, larger, prospective, high-quality studies with longer follow-up are needed to standardize the technical aspects of the procedure, identify the optimal injectate, determine the ideal frequency of repeat treatments, if required, and potentially compare GON hydrodissection with cohort or usual treatment controls such as GON blocks without hydrodissection using lidocaine or lidocaine plus steroid.

Overall, ultrasound-guided GON hydrodissection with D5W represents a minimally invasive, potentially effective, option with inherently favorable safety due to the avoidance of systemic effects. Its place in the treatment algorithm of occipital neuralgia, either as a supplement to current approaches, a stand-alone treatment, or an alternative to more invasive approaches, is dependent on future research outcomes.

## Figures and Tables

**Figure 1 diagnostics-14-01380-f001:**
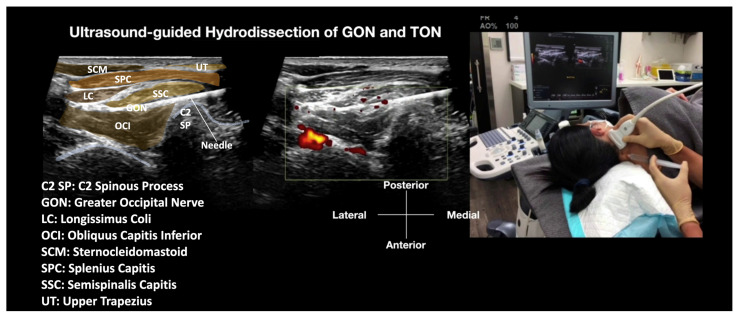
Ultrasound-guided hydrodissection of the greater occipital nerve (GON) between the semispinalis capitis (SSC) and obliquus capitis inferior (OCI) muscles, with a medial to lateral and posterior to anterior approach. The patient is positioned in the left lateral decubitus position, and the treatment side (right side) is positioned upwards, allowing the ultrasound probe to be placed laterally to visualize the fascial plane between the SSC and OCI muscles. The needle is then advanced from the posteromedial to the anterolateral aspect to hydrodissect the GON both above and below the GON [25].

**Figure 2 diagnostics-14-01380-f002:**
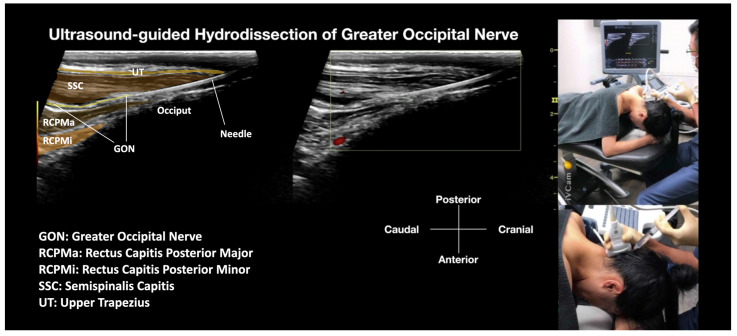
Cranial to Caudal Approach of Ultrasound-Guided Hydrodissection of the Greater Occipital Nerve (GON) within the Semispinalis Capitis (SSC) and Upper Trapezius (UT) Muscles. The procedural setup, the needle insertion, and the path of the hydrodissection procedure within the SSC and UT muscles have been clearly shown. The procedure aims to target the region where the GON passes through and potentially becomes entrapped, providing a visual representation of the intervention [25].

## Data Availability

Data related to this study has been included in the manuscript.

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
