# Peer review of "Novel Ultrasound-Guided Hydrodissection with 5% Dextrose for the Treatment of Occipital Neuralgia Targeting the Greater Occipital Nerve"

_diagnostics, 2024, doi:10.3390/diagnostics14131380_

Round 1
Reviewer 1 Report
Comments and Suggestions for Authors
GENERAL COMMENTS
This is the reporting of 2 cases evaluating the effects of ultrasound-guided hydrodissection of the greater occipital nerve using 5% dextrose in the pain management of 2 patients with occipital neuralgia. Overall, the article is interesting and provides details regarding techniques and management. I think, nonetheless, that authors should use more contemporary pain related terms, e.g., irritated or compressed nerve is somewhat outdated, and there is many information that lacks literature support which I’ll detail later. Being Diagnostics a scholarly journal on medical diagnosis I miss the clinical reasoning, including differentiation/decision-making process, particularly in the 2nd case, of choosing another site for the injection or dose. Authors provide the anatomical substrate, but I’m wonder if was something in the clinic (e.g., pain location) or in the ultrasound imaging that made the authors hypothesized whether the new site was a good or better option for injection, or was it a trial-error method? Considering the potential side-effects of the procedure, as you outline in lines 346–371, what are your thoughts on that trial-error method (if this was the case).
SPECIFIC COMMENTS
Abstract
Line 29: remove “reports”, or state, “Two cases are reported”, in alternative.
Line 34: I suggest removing “quality of life” since you haven’t used any health-related quality-of-life outcome measure. The Neck Disability Index does not contain a quality-of-life domain or dimension.
Introduction
Line 56: Physical therapy is a profession not an intervention. Inform if the physical therapists’ intervention was exercise, or TENS, or manual therapy, etc.
The first 2 paragraphs don’t have any citation/reference. Where did you get this information? Remember that Diagnostics is a scholarly journal following high standards on the reporting. Additionally, what about surgery? It can also provide partial and/or temporary relief? What are systematic reviews saying on the effectiveness of conservative and of surgical interventions and their shortcomings?
Line 75: remove the “s” on “detail”.
Case presentation
Line 99: How did you assessed “concentration” so you could say it improved immediately?
Line 119: I believe “pain” should not be between neck and disability (NDI). Or is this another PROM?
Line 128: “has” instead of “was”?
Discussion
Lines 300–301: “of the scalp”, and actually, GON does not send pain signals, nor it send them to the brain. It sends nociceptive signals (among other sensory modalities) to the dorsal horn of the spinal cord. Please amend.
Line 301: Consider using a less cause-effect language because
compression alone may cause, mostly, a reduction in somatosensory function of nerves not an augmentation. I would go for something more neutral and a less categorical mechanism. Something like: When the nerve itself and/or the nerve bed are sensitized due to chemical (e.g., inflammatory substances) or mechanical (scar tissue) stimulus it may cause pain and tenderness in the area.
Line 302–304: This sentence should also follow a less definitive mechanistic language of cause and effect.
Line 335–338: Is this based on available literature and your experience, or both? This appears not to be the circumstance of the 2 patients reported in here. Please clarify readers (and reviewers).
Lines 346–371:
Where is this coming for? The literature? Provide the citations/references.
Conclusions
Line 386: I suggest replacing “evidence is” by “results are”.
Comments on the Quality of English LanguageI provide examples in my Comments and Suggestions for Authors.
Author Response
Reviewer 1
GENERAL COMMENTS
This is the reporting of 2 cases evaluating the effects of ultrasound-guided hydrodissection of the greater occipital nerve using 5% dextrose in the pain management of 2 patients with occipital neuralgia. Overall, the article is interesting and provides details regarding techniques and management. I think, nonetheless, that authors should use more contemporary pain related terms, e.g., irritated or compressed nerve is somewhat outdated, and there is many information that lacks literature support which I’ll detail later. Being Diagnostics a scholarly journal on medical diagnosis I miss the clinical reasoning, including differentiation/decision-making process, particularly in the 2nd case, of choosing another site for the injection or dose. Authors provide the anatomical substrate, but I’m wonder if was something in the clinic (e.g., pain location) or in the ultrasound imaging that made the authors hypothesized whether the new site was a good or better option for injection, or was it a trial-error method? Considering the potential side-effects of the procedure, as you outline in lines 346–371, what are your thoughts on that trial-error method (if this was the case).
Response, Thank you for your comments. We have added the diagnostic criteria for occipital neuralgia and differential diagnoses in the introduction. Line 50 to line 69, “Diagnosis of occipital neuralgia requires severe, episodic, and sharp, shooting, or stab-bing pain in one or more occipital nerve distributions, episodes that last only minutes, and relief after local anesthetic block[2]. Allodynia or dysesthesia may be present in the nerve distribution [3]. A Tinel’s sign is commonly found [4]. A recent study of 800 patients who presented with a chief complaint of headache found 25% met criteria for occipital neu-ralgia[5]. It is important to note, however, that up to 85% of patients with occipital neu-ralgia may have an additional headache type[5]. For that reason, accompanying tension, migraine, and cluster headaches, other neuralgia, or mechanical neck pain causing re-ferred to the head should be kept in mind for treatment consideration as well, to maximize overall functional impact of treatment[6].”
We have also listed the patient selection criteria in the technical notes for each technique.
Lines 329 – 341: Patient selection: Patients fulfill the diagnostic criteria as described in the introduction for occipital neuralgia, with other differential diagnoses excluded. They have failed other conservative treatments. The local tenderness is felt mainly over the OCI (usually slightly lateral than the GON entrapped within the SCC and UT group), and compression may reproduce their occipital neuralgia-type pain between the SCC and OCI, but not within the SCC and UT. There is no swelling and tenderness of the SCC and UT. Ultrasound shows swelling of the GON compared to the normal, non-painful side between the SCC and OCI, or a cross-sectional area exceeding the upper limit of 3 mm2[27]. If they have swelling and tenderness of the SCC and UT (more medial than the entrapment between the SCC and OCI group), it means the GON may also be entrapped within the SCC and UT, indicating that the GON may also need to be hydrodissected using the cranial to caudal technique described below.
Lines 490 – line 505; “Patient selection: Patients fulfill the diagnostic criteria as described in the introduc-tion for occipital neuralgia, with other differential diagnoses excluded. They have failed other conservative treatments. Local tenderness and swelling are felt mainly over the UT and SCC (usually more medial than the GON entrapped between the SCC and OCI group). Compression of the SCC and UT may reproduce their occipital neuralgia-type pain, but not compression between the SCC and OCI, and there is no swelling and tenderness between the SCC and OCI. Ultrasound may show swelling of the GON as a hypoechoic structure passing through the SCC and UT, compared to the normal, non-painful side, or a cross-sectional area exceeding the upper limit of 3 mm2. However, the cross-sectional area of the GON within the SCC and UT is very difficult to measure accurately due to the similarity of echogenicity with the sur-rounding muscles. If they have swelling and tenderness of the SCC and OCI (more lateral than the entrapment within the SCC and UT group), it suggests an additional entrapment site of the GON between the SCC and OCI, indicating a need to add hydrodissection using the posteromedial to anterolateral approach technique de-scribed above (in 3.1)."
We have updated the history of patient in case 2, lines 263 to 266, “ She described the pain as a baseline dull ache and tightness with frequent exacerbation to severe, episodic, and sharp, shooting, or stabbing pain in the right GON distributions, episodes that last only minutes, and relieved temporarily after local anesthetic block.” Lines 276 to 278 “Two courses of TENS, manual therapy, magnetic and infrared therapy by the physiotherapist did not provide any relief for the patient (from toat. “The patient in case 2 had previously undergone an ultrasound-guided GON block between the SCC and OCI by another pain physician, which only provided temporary pain relief due to the blocking effect on the GON. However, the occipital neuralgia persisted. We mentioned that, and explaining that we performed a different approach to address the true GON entrapment within the SCC and UT. We modified lines 280 to 292, “Together with the swelling and tenderness of the SCC and UT, where the GON passes through these muscles, and considering the limited success of the previous interventions, she provided informed consent to undergo an ultrasound-guided hydrodissection using 20 mL of D5W without local anesthetic, of the long-axis of the GON within the SSC and UT as well as between the SSC and RCPMa [25] from cranial to caudal approach (Figure 2 and Video 2[25])”
SPECIFIC COMMENTS
Abstract
- Line 29: remove “reports”, or state, “Two cases are reported”, in alternative.
Response: Thank you for your comment. Changes were made in line 31, “Two cases are reported.”
- Line 34: I suggest removing “quality of life” since you haven’t used any health-related quality-of-life outcome measure. The Neck Disability Index does not contain a quality-of-life domain or dimension.
Response: Thank you for your comment. Line 36, “Quality of life” was removed.
Introduction
- Line 56: Physical therapy is a profession not an intervention. Inform if the physical therapists’ intervention was exercise, or TENS, or manual therapy, etc.
Response: Thank you for your comment. Line 66 “Postural exercise, manual therapy, or modalities such as TENS” were substituted.
- The first 2 paragraphs don’t have any citation/reference. Where did you get this information? Remember that Diagnostics is a scholarly journal following high standards on the reporting. Additionally, what about surgery? It can also provide partial and/or temporary relief? What are systematic reviews saying on the effectiveness of conservative and of surgical interventions and their shortcomings?
Response: Thank you for your comment. Citations have been added to the first 3 paragraphs of the introduction along with modification of the structure of the introduction. Swanson et als. systematic review and meta-analysis was included. The surgical option was mentioned, along with the limited outcome literature available at this time. Line 44 to line 97 “Occipital neuralgia is a common and debilitating condition characterized by pain and tenderness in the suboccipital region, with referral to the head along the distribution of the greater, lesser, or third occipital nerves[1]. This type of headache can be caused by a variety of factors, including muscle tension, joint inflammation, and nerve entrapment. Diagnosis of occipital neuralgia requires severe, episodic, and sharp, shooting, or stab-bing pain in one or more occipital nerve distributions, episodes that last only minutes, and relief after local anesthetic block[2]. Allodynia or dysesthesia may be present in the nerve distribution [3]. A Tinel’s sign is commonly found [4]. A recent study of 800 patients who presented with a chief complaint of headache found 25% met criteria for occipital neu-ralgia[5]. It is important to note, however, that up to 85% of patients with occipital neu-ralgia may have an additional headache type[5]. For that reason, accompanying tension, migraine, and cluster headaches, other neuralgia, or mechanical neck pain causing re-ferred to the head should be kept in mind for treatment consideration as well, to maximize overall functional impact of treatment[6].
The traditional treatment of occipital neuralgia typically involves a combination of treatments including pharmacological and nonpharmacological managements e.g. non-steroidal anti-inflammatory drugs (NSAIDs), muscle relaxants, opioids, tricyclic antide-pressants, postural exercise, manual therapy, or modalities such as TENS[7]. Perineural injection (blocks) using local anesthetics or corticosteroids have been extensively utilized. A recent meta-analysis of nerve block efficacy for occipital headache compared to baseline status, and compared to other treatments was performed by Evans et al[8]. They reported significant improvement in frequency and severity of occipital headaches compared to baseline for 6 weeks and 6 months respectively, with a mean improvement in pain se-verity of 40-45%. In active treatment comparison trials, cryoneurolysis, pulsed radiofre-quency treatment, and GON treatment with botulinum toxin A (BTX) outperformed anesthetic blocks at 6 months for magnitude of pain improvement[8]. Among ablation techniques, pulsed radiofrequency or cryoablation preserve nerve architecture better than thermal radiofrequency or chemical ablation, with less potential for hypesthesia or dysesthesia[6], but long term follow-up studies are lacking[9]. BTX injection has a better side effect profile, but requires repetition indefinitely[6]. Surgical approaches such as partial resection of the obliquus capitis or C2 gangliotomy have had success, but carry the risk of neuromas and causalgia[6]. Two studies have been reported with favorable results and minor side effects, but the retrospective nature of these data collections studies, with inherent bias, selective recall, incomplete data capture, and a lack of formal side effect analysis, limit the strength of conclusions[10, 11].
Occipital neuralgia almost always results from compression of the greater occipital nerve (GON), lesser occipital nerve, or third occipital nerve at one or more points along their course, and the GON is the primary source in 90% of case[12, 13]. The GON is the dorsal rami of the second cervical nerve root. After branching out from the C2 nerve root, it wraps around the obliquus capitis inferior (OCI) muscle, ascends between the OCI and semispinalis capitis (SSC) muscle, and then between the rectus capitis posterior major (RCPMa) muscle and SSC. Then it pierces through the SSC and upper trapezius (UT) to surface to the subcutaneous tissues in the occipital area[14]. Consistent with the compli-cated course of the GON, technique appears to play a significant role in GON block outcomes. Using ultrasonographic or fluoroscopic guidance to block the GON in the suboccipital compartment enhances accuracy of injection[15] and durability of benefit[16, 17] compared to classical technique for GON block at the superior nuchal line. A guided suboccipital compartment approach has been recommended for future comparative studies between GON block and Botulinum toxin injection or neurolysis[8].”
- Line 75: remove the “s” on “detail”.
Response: Thank you for your comment. Line 127, the “s” on “details” has been removed.
Case presentation
- Line 99: How did you assess “concentration” so you could say it improved immediately?
Response: Thank you for your comment. The concentration span of the patient was assessed by the digit span test for short-term memory. Lines150 tp 152 “The digit span test for short-term memory revealed a forward recall of 5 digits and a backward recall of 4 digits[24], below normal limits.” Post injection, line 160-162, “..and her concentration span improved as noted by a forward recall of 8 digits and a backward recall of 6 digits on the digit span test for short-term memory[24]”
- Line 119: I believe “pain” should not be between neck and disability (NDI). Or is this another PROM?
Response: Thank you for your comment. Pain has been removed and abbreviation NDI has been used consistently.
- Line 128: “has” instead of “was”?
Response: Thank you for your comment, “she was provided informed consent to undergo” was changed to line 154, “She provided informed consent to undergo”.
Discussion
- Lines 300–301: “of the scalp”, and actually, GON does not send pain signals, nor it send them to the brain. It sends nociceptive signals (among other sensory modalities) to the dorsal horn of the spinal cord. Please amend.
Response: Thank you for your comment. The sentence related to the GON has been altered to lines 679 to 680 “ The GON is a sensory nerve that sends nociceptive signals, among other sensory modalities, to the dorsal horn of the spinal cord.”
- Line 301: Consider using a less cause-effect language because compression alone may cause, mostly, a reduction in somatosensory function of nerves not an augmentation. I would go for something more neutral and a less categorical mechanism. Something like: When the nerve itself and/or the nerve bed are sensitized due to chemical (e.g., inflammatory substances) or mechanical (scar tissue) stimulus it may cause pain and tenderness in the area.
Response: Thank you for your comment. The sentence was altered to lines 680 to 682“When the nerve itself and/or the nerve bed are sensitized due to chemical (e.g. inflammatory substances) or mechanical (scar tissue) stimuli, it may cause pain and tenderness in the suboccipital region.”
- Line 302–304: This sentence should also follow a less definitive mechanistic language of cause and effect.
Response: Thank you for your comment. The statement is question was redundant with the introductory, multiply cited, sentence of that paragraph, and was deleted
- Line 335–338: Is this based on available literature and your experience, or both? This appears not to be the circumstance of the 2 patients reported in here. Please clarify readers (and reviewers).
Response: Thank you for your comment. This section was clarified to state lines 811 to 812“Empirical observations suggest that the response to hydrodissection can vary significantly between individuals, and large-scale studies are needed to standardize this technique, its protocol, and indications for this procedure”. A previous citation was provided in which this technique was mentioned in a letter to the editor, reference number 25.
- Lines 346–371: Where is this coming from? The literature? Provide the citations/references.
Response: Thank you for your comment. The potential side effects of the ultrasound-guided GON hydrodissection described in this manuscript are primarily related to the needling procedure itself. Systemic effects or risks are generally minimized because this technique typically involves use of D5W without local anesthetic. So we reformulated the side effect session in the end of the discussion session. Lines 821 to 833: • Possible Side Effects of Ultrasound-Guided GON Hydrodissection using D5W without local anesthetic:
- Potential side effects of the ultrasound-guided GON hydrodissection described in this manuscript according to authors’ experiences are primarily related to the needling procedure itself. Systemic effects or side effects from unintentional flow of fluid to surrounding structures are not expected using D5W as the in-jectate rather than a local anesthetic.
- Its side effects profile is similar to, but considerably less severe than, the ultrasound-guided GON block. This is because a hydrodissecting technique is used which theoretically pushes the nerves and vascular structures away from the needle tip, further reducing the chance of inadvertant vascular puncture and nerve irritation.
- Temporary discomfort may be possible due to pressure effect of the soloution.”
Conclusions
- Line 386: I suggest replacing “evidence is” by “results are”.
Response: Thank you for your comment. “results are” was substituted for “evidence is”. Line 841” While retrospective results are promising, larger, prospective, high-quality studies…”

Reviewer 2 Report
Comments and Suggestions for Authors
Here are the review suggestions for the manuscript titled "Novel Ultrasound-Guided Hydrodissection with 5% Dextrose Techniques for the Treatment of Occipital Neuralgia Targeting the Greater Occipital Nerve":
1. Provide a more detailed comparative analysis of the proposed techniques with other existing treatments for occipital neuralgia, highlighting the unique contributions and potential advantages of the ultrasound-guided hydrodissection with 5% dextrose.
2. Methodology: Include more technical details about the ultrasound-guided hydrodissection procedure, such as the type of ultrasound machine used, specific settings, and step-by-step procedural guidelines. This will help other practitioners replicate the study.
3. Selection Criteria: Clarify the criteria used for selecting the patients included in the case reports. This will provide a better understanding of the patient population that might benefit most from these techniques.
4. Results Presentation: Include more quantitative data to support the findings. For instance, provide pre- and post-procedure pain scores, functional improvement metrics, and statistical analysis where applicable.
5. Discussion: Provide a more detailed explanation of the proposed mechanisms by which the 5% dextrose solution alleviates pain and improves function. Reference relevant physiological and biochemical studies to support these explanations.
Author Response
Reviewer 2:
Here are the review suggestions for the manuscript titled "Novel Ultrasound-Guided Hydrodissection with 5% Dextrose Techniques for the Treatment of Occipital Neuralgia Targeting the Greater Occipital Nerve":
- Provide a more detailed comparative analysis of the proposed techniques with other existing treatments for occipital neuralgia, highlighting the unique contributions and potential advantages of the ultrasound-guided hydrodissection with 5% dextrose.
Response: Thank you for your comment. A cited and succinct comparative analysis of available techniques was summarized in the introduction. The reason for use of D5W was alluded to with two citations in the introduction, but was covered in more detail in the discussion of potential therapeutic mechanisms. Lines 125 to 132, “We have observed some patients who cannot lie prone. Additionally, the GON en-trapment may be inside the SSC and UT, when the nerve pierces through these two muscles. Herein we describe two different approaches in detail that are appropriate for general use, and feasible for use with prone positioning difficulty or entrapment inside SSC and UT, elaborating their advantages and disadvantages. D5W was utilized as the injectate in these two cases, rather than anesthetic injection. This is related to an apparent therapeutic effect of dextrose itself [21], beyond that of a mechanical effect of injection[22], which will be addressed in the discussion section.
Lines 687 to 762, “3.4. Mechanism of Action of Glucose:
The evidence for a direct ameliorative effect of dextrose on neurogenic pain is ac-cumulating in three primary areas. One proposed neurogenic mechanism is the “gly-copenic hypothesis”. Kim et al. found dextrose superior to saline or 0.5% lidocaine for trigger injection and proposed an “energy benefit” from use of dextrose 12.5% in the injectate[37]. MacIver et al. demonstrated that C fibers exposed to removal of D-glucose (dextrose) fire 650% faster within 20 minutes and return to a normal firing rate with reintroduction of dextrose[38]. Dr John Lyftogt proposed that high-energy-requiring C fibers use up their available energy stores through enhanced firing rates in the presence of chronic pain, resulting in a perineural glycopenic state[39]. As glycolysis is the primary source of ATP production[40], nerves may have insufficient ATP to power their sodium potassium pump, which is critically required for repolarization. Sensory nerves that do not reach a minus 70 millivolt transmembrane potential will fire much more easily and can fire constantly, perpetuating chronic pain[41]. Although dextrose upon injection only donates 2 ATP initially via the first glycolytic step, that occurs in seconds, and may explain the immediate benefit of dextrose injection.
Multiple observations have suggested that dextrose increases the level of substance P (SP). Although substance P (SP) has been characterized as promoting nociception[42], that effect appears to be confined to the spinal cord[43]. SP reduced hyperalgesia and allodynia via supraspinal injection in a rat inflammatory pain model[44], reduced nociception in an acid-induced pain model[45], and improved neuropathic pain after intravenous admin-istration in a neuropathic pain model[46]. SP activates opioid receptors via NK1 receptor binding[47], and is being considered a candidate for neuropathic pain treatment in human trials[46]. Han et showed that dextrose likely exerts its analgesic effect by inducing the release of substance P through the activation of the acid-sensing ion channel 1a (ASIC1a)[48].Topol et al. demonstrated analgesia after injection of dextrose in painful osteoarthritic knees, reducing pain with walking by 4.2 on the 0-10 NRS within 20 minutes, accompanied by a rise in synovial fluid SP by 112% at one week[49].
Other studies have demonstrated an anti-inflammatory effect of dextrose on nerves. An in vitro study by Wu et al. showed that dextrose decreased the inflammatory markers IL-6 and IL1B in TNF-a exposed nerve cells in a dose-dependent manner[33]. Similarly, Cherng et al. found that TNF-a-exposed nerve cells exhibited increased survival and decreased reactive oxygen species (ROS) in response to increasing concentrations of dextrose[31]. Topol et al. showed that medium-term (3 month) levels of the important degenerative neuropeptide Y (NPY) dropped markedly in synovial fluid in human stage IV osteoarthritic knees[49] after monthly dextrose injection.
Consistent with potential dextrose effects on C fiber energetics, SP levels, inflam-matory markers, or other mechanisms as yet unexplored, we propose that dextrose has a multifaceted mechanism by which it alleviates pain and reduces inflammation. The role of dextrose in modulating key neuropeptides and ion channels presents exciting avenues for further research and clinical applications.
- Methodology: Include more technical details about the ultrasound-guided hydrodissection procedure, such as the type of ultrasound machine used, specific settings, and step-by-step procedural guidelines. This will help other practitioners replicate the study.
Response: Thank you for your comment. Detailed descriptions of the patient selection, patient position, ultrasound machine and probe, and identification of important sonoanatomy, needle, injectate and method are included in the technical note of each method, step-by-step instructions on how to perform these techniques were added in the technical notes section of the manuscript. Lines 332 to 384, “• Patient selection: Patients fulfill the diagnostic criteria as described in the introduction for occipital neuralgia, with other differential diagnoses excluded. They have failed other conservative treatments. The local tenderness is felt mainly over the OCI (usually slightly lateral than the GON entrapped within the SCC and UT group), and compression may reproduce their occipital neuralgia-type pain between the SCC and OCI, but not within the SCC and UT. There is no swelling and tenderness of the SCC and UT. Ultrasound shows swelling of the GON compared to the normal, non-painful side between the SCC and OCI, or a cross-sectional area exceeding the upper limit of 3 mm2[27]. If they have swelling and tenderness of the SCC and UT (more medial than the entrapment between the SCC and OCI group), it means the GON may also be entrapped within the SCC and UT, indicating that the GON may also need to be hydrodissected using the cranial to caudal technique described below.
- Patient's Position: The patient is typically positioned in a lateral decubitus position with the GON under treatment facing upward. This positioning allows for an al-ternative treatment option if the patient is unable to lie prone. From the patient's perspective, it is generally more comfortable if the needle comes from behind, es-pecially when they are fully awake, as it prevents them from seeing the needle.
- The Ultrasound Machine and Probe: The ultrasound machine is placed in front of the patient facing the treating physician to optimize ergonomics. The ultrasound used in this manuscript was the Logiq S7 (General Electric, Boston, MA, USA). A linear transducer with a broad frequency range (GE L3-12 D, General Electric, Boston, MA, USA) is used, which provides sufficient penetration and resolution. It is positioned on the transverse plane, aligned with the long axis of the OCI, emitting ultrasound waves to visualize the underlying anatomy.
- Identifications of Important Sonoanatomy: Obliquus Capitis Inferior Muscle: This is a striated muscle that originates from the spinous process of the axis (C2) and inserts into the posterior aspect of the transverse process of the atlas (C1) (Figure 1). Semispinalis Capitis Muscle: The SSC muscle is a deep neck muscle visualized in the short axis view. It is located superficial to the OCI and underneath the splenius capitis and UT. The UT and sternocleidomastoid muscles, being the most superficial muscles in that region (Figure 1). Greater Occipital Nerve: The GON is identified as a hypoechoic oval structure situated between the SSC and OCI (Figure 1). In cases where there is swelling or inflammation, the GON may appear enlarged compared to the normal side.
- Needle: A 25G or 27G 2-inch hypodermic needle is typically used. It is inserted under ultrasound guidance, approaching the GON for the hydrodissection procedure both above and below the GON (Figure 1 and Video 1[25]).
- Injectate: D5W without local anesthetic.
- Method: Appropriate disinfection preparation uses 2% chlorhexidine in 75% isopropyl alcohol. It is also advisable to use the same disinfectant solution as the contact media for the ultrasound transducer[28]. The needle entry point is anesthetized with 0.5% lidocaine to create a skin wheel. With ultrasound guidance, a 25G or 27G 2-inch hypodermic needle is inserted using an in-plane technique. The needle tip is ad-vanced to the site of the GON using a hydrodissection technique. Injection of D5W during needle advancement pushes away any soft tissue in front of the needle tip and creates a halo, allowing the needle tip to follow from the subcutaneous fat to the site and the whole tract of the GON to be hydrodissected. The needle is guided to hy-drodissect the GON between the SSC and OCI, and the soft tissue is completely hydrodissected above and below the GON, with the GON completely encircled with anechoic fluid. Typically, 20 to 30 mL of D5W without local anesthetic will be used for one side of GON hydrodissection, from skin entry to complete hydrodissection of the GON, resulting in the patient's pain being relieved.”
And lines 493 to 547, “• Patient selection: Patients fulfill the diagnostic criteria as described in the introduc-tion for occipital neuralgia, with other differential diagnoses excluded. They have failed other conservative treatments. Local tenderness and swelling are felt mainly over the UT and SCC (usually more medial than the GON entrapped between the SCC and OCI group). Compression of the SCC and UT may reproduce their occipital neuralgia-type pain, but not compression between the SCC and OCI, and there is no swelling and tenderness between the SCC and OCI. Ultrasound may show swelling of the GON as a hypoechoic structure passing through the SCC and UT, compared to the normal, non-painful side, or a cross-sectional area exceeding the upper limit of 3 mm2. However, the cross-sectional area of the GON within the SCC and UT is very difficult to measure accurately due to the similarity of echogenicity with the sur-rounding muscles. If they have swelling and tenderness of the SCC and OCI (more lateral than the entrapment within the SCC and UT group), it suggests an additional entrapment site of the GON between the SCC and OCI, indicating a need to add hydrodissection using the posteromedial to anterolateral approach technique de-scribed above (in 3.1).
- Patient's Position: The patient is typically positioned in a prone position with the head and neck fully flexed to allow for optimal ultrasound scanning and guided injection. This position also helps to minimize the curvature in the suboccipital re-gion, which can otherwise make the procedure challenging to perform.
- The Ultrasound Machine and Probe: The ultrasound machine is placed by the side of the treatment table with the monitor in front of the treating physician for optimal ergonomics. The ultrasound used in this manuscript was the Logiq S7 (General Electric, Boston, MA, USA). A linear transducer with a broad frequency range (GE L3-12 D, General Electric, Boston, MA, USA) was used, which provided sufficient penetration and resolution. The transducer was positioned in the sagittal plane, aligned with the long axis of the SSC and UT muscles.
- Identifications of Important Sonoanatomy: Upper Trapezius Muscle: The UT muscle is located superficially in this region, beneath the adipose tissue and fascia. It is a thin layer of striated muscle. Semispinalis Capitis Muscle: The SSC muscle is a deep neck muscle that is visualized in a long-axis view. It is located superficial to the rectus capitis superior muscle and underneath the UT and splenius capitis muscles. Greater Occipital Nerve: The GON is identified as a hypoechoic fascicular structure that passes through the SSC and UT muscles. In cases where there is swelling or in-flammation, the GON may appear enlarged compared to the normal side.
- Needle: A 25G or 27G 2-inch hypodermic needle is inserted under ultrasound guidance to access the GON for hydrodissection above and below the nerve.
- Injectate: D5W without local anesthetic.
- Method: Because of the presence of hair in this region, a larger amount of 2% chlorhexidine in isopropyl alcohol is advised to be used as the disinfection agent. The disinfectant solution may need to be applied several times to ensure proper disin-fection. It is also advised to use the same disinfectant solution as the contact media for the ultrasound transducer. The needle entry point is anesthetized with 0.5% lidocaine to create a skin wheel. With ultrasound guidance, a 25G or 27G 2-inch hypodermic needle is inserted. The physician keeps injecting with D5W with needle advancement to push away any soft tissue in front of the needle tip and create a halo, allowing the needle tip to follow, from the subcutaneous fat to the site. This allows for the whole tract of the GON to be hydrodissected, within the UT and SCC, and between the SCC and RCPMa, as the GON passes through these anatomical structures. The soft tissue is completely hydrodissected above and below, lateral, and medial to the GON, with the GON completely encircled with anechoic fluid. During the procedure, in-plane and out-of-plane techniques will be used interchangeably to validate the needle position. Typically, 20 to 30 mL of D5W without local anesthetic will be used for one side of GON hydrodissection, from skin entry to complete hydrodissection of the GON, resulting in the patient's pain being relieved.”
- Selection Criteria: Clarify the criteria used for selecting the patients included in the case reports. This will provide a better understanding of the patient population that might benefit most from these techniques.
Response: Thank you for your comment. The diagnostic criteria of patients with occipital neuralgia were described in the introduction, lines 53 to 62 “Diagnosis of occipital neuralgia requires severe, episodic, and sharp, shooting, or stabbing pain in one or more occipital nerve distributions, episodes that last only minutes, and relief after local anesthetic block[2]. Allodynia or dysesthesia may be present in the nerve distribution [3]. A Tinel’s sign is commonly found [4]. A recent study of 800 patients who presented with a chief complaint of headache found 25% met criteria for occipital neuralgia[5]. It is important to note, however, that up to 85% of patients with occipital neuralgia may have an additional headache type[5]. For that reason, accompanying tension, migraine, and cluster headaches, other neuralgia, or mechanical neck pain causing referred to the head should be kept in mind for treatment consideration as well, to maximize overall functional impact of treatment[6].
The “patient selection”: section was added in the technical note of each technique. Lines 332 to 344, “• Patient selection: Patients fulfill the diagnostic criteria as described in the introduction for occipital neuralgia, with other differential diagnoses excluded. They have failed other conservative treatments. The local tenderness is felt mainly over the OCI (usually slightly lateral than the GON entrapped within the SCC and UT group), and compression may reproduce their occipital neuralgia-type pain between the SCC and OCI, but not within the SCC and UT. There is no swelling and tenderness of the SCC and UT. Ultrasound shows swelling of the GON compared to the normal, non-painful side between the SCC and OCI, or a cross-sectional area exceeding the upper limit of 3 mm2[27]. If they have swelling and tenderness of the SCC and UT (more medial than the entrapment between the SCC and OCI group), it means the GON may also be entrapped within the SCC and UT, indicating that the GON may also need to be hydrodissected using the cranial to caudal technique described below.”
And Lines 493 to 508, “• Patient selection: Patients fulfill the diagnostic criteria as described in the introduc-tion for occipital neuralgia, with other differential diagnoses excluded. They have failed other conservative treatments. Local tenderness and swelling are felt mainly over the UT and SCC (usually more medial than the GON entrapped between the SCC and OCI group). Compression of the SCC and UT may reproduce their occipital neuralgia-type pain, but not compression between the SCC and OCI, and there is no swelling and tenderness between the SCC and OCI. Ultrasound may show swelling of the GON as a hypoechoic structure passing through the SCC and UT, compared to the normal, non-painful side, or a cross-sectional area exceeding the upper limit of 3 mm2. However, the cross-sectional area of the GON within the SCC and UT is very difficult to measure accurately due to the similarity of echogenicity with the sur-rounding muscles. If they have swelling and tenderness of the SCC and OCI (more lateral than the entrapment within the SCC and UT group), it suggests an additional entrapment site of the GON between the SCC and OCI, indicating a need to add hydrodissection using the posteromedial to anterolateral approach technique de-scribed above (in 3.1).”
- Results Presentation: Include more quantitative data to support the findings. For instance, provide pre- and post-procedure pain scores, functional improvement metrics, and statistical analysis where applicable.
Response: Thank you for your comment. We have provided time 0 and 6 month 0-10 NRS pain and NCI scores, and have commented on the magnitude of change in the NDI compared to its minimal clinically important difference per cited literature. In Case 1, Lines 159 to 162, “The patient's pain dropped from 8/10 to 0/10 immediately after the hydrodissection and her concentration span improved as noted by a forward recall of 8 digits and a backward recall of 6 digits on the digit span test for short-term memory[24].” Line 164 to 165,” Her pain score sustained at 0-1/10 at 6 months and her NDI at 6 months was 4/50, consistent with no disability.” Line 166 to 168,” Given an MCID for the NDI in those without radicular upper extremity pain of 5.5 points, this would be approximate 2.5 MCID levels of change, or an 80% improvement[26].”
In Case 2, line 296,” Upon completion of the procedure, the NRS dropped from 9/10 to 1/10,…” Lines 299 to 301,” The NRS sustained at 0-1/10, with an NDI of 3/50 at 6 months, and instructions to re-contact with pain recurrence. This improvement in the NDI represents 3.8 times the MCID, or 85%.”
- Discussion: Provide a more detailed explanation of the proposed mechanisms by which the 5% dextrose solution alleviates pain and improves function. Reference relevant physiological and biochemical studies to support these explanations.
Response: Thank you for your comment. Under section 3.4, we have expanded the potential mechanism discussion, with appropriate citations. Lines 687 to 762, “3.4. Mechanism of Action of Glucose:
The evidence for a direct ameliorative effect of dextrose on neurogenic pain is ac-cumulating in three primary areas. One proposed neurogenic mechanism is the “gly-copenic hypothesis”. Kim et al. found dextrose superior to saline or 0.5% lidocaine for trigger injection and proposed an “energy benefit” from use of dextrose 12.5% in the injectate[37]. MacIver et al. demonstrated that C fibers exposed to removal of D-glucose (dextrose) fire 650% faster within 20 minutes and return to a normal firing rate with reintroduction of dextrose[38]. Dr John Lyftogt proposed that high-energy-requiring C fibers use up their available energy stores through enhanced firing rates in the presence of chronic pain, resulting in a perineural glycopenic state[39]. As glycolysis is the primary source of ATP production[40], nerves may have insufficient ATP to power their sodium potassium pump, which is critically required for repolarization. Sensory nerves that do not reach a minus 70 millivolt transmembrane potential will fire much more easily and can fire constantly, perpetuating chronic pain[41]. Although dextrose upon injection only donates 2 ATP initially via the first glycolytic step, that occurs in seconds, and may explain the immediate benefit of dextrose injection.
Multiple observations have suggested that dextrose increases the level of substance P (SP). Although substance P (SP) has been characterized as promoting nociception[42], that effect appears to be confined to the spinal cord[43]. SP reduced hyperalgesia and allodynia via supraspinal injection in a rat inflammatory pain model[44], reduced nociception in an acid-induced pain model[45], and improved neuropathic pain after intravenous admin-istration in a neuropathic pain model[46]. SP activates opioid receptors via NK1 receptor binding[47], and is being considered a candidate for neuropathic pain treatment in human trials[46]. Han et showed that dextrose likely exerts its analgesic effect by inducing the release of substance P through the activation of the acid-sensing ion channel 1a (ASIC1a)[48].Topol et al. demonstrated analgesia after injection of dextrose in painful osteoarthritic knees, reducing pain with walking by 4.2 on the 0-10 NRS within 20 minutes, accompanied by a rise in synovial fluid SP by 112% at one week[49].
Other studies have demonstrated an anti-inflammatory effect of dextrose on nerves. An in vitro study by Wu et al. showed that dextrose decreased the inflammatory markers IL-6 and IL1B in TNF-a exposed nerve cells in a dose-dependent manner[33]. Similarly, Cherng et al. found that TNF-a-exposed nerve cells exhibited increased survival and decreased reactive oxygen species (ROS) in response to increasing concentrations of dextrose[31]. Topol et al. showed that medium-term (3 month) levels of the important degenerative neuropeptide Y (NPY) dropped markedly in synovial fluid in human stage IV osteoarthritic knees[49] after monthly dextrose injection.
Consistent with potential dextrose effects on C fiber energetics, SP levels, inflam-matory markers, or other mechanisms as yet unexplored, we propose that dextrose has a multifaceted mechanism by which it alleviates pain and reduces inflammation. The role of dextrose in modulating key neuropeptides and ion channels presents exciting avenues for further research and clinical applications.

Round 2
Reviewer 1 Report
Comments and Suggestions for Authors
I have read attentively authors responses and amendments and would like to congratulate them on their work. This version has reached high quality on the reporting and discussion/decision-making process of the two cases.
A couple of minor amendments as final recommendation:
– You have not defined minimal clinical important difference upon first appearance in the text. Many readers may not recognize this clinically relevant statistical measure to evaluate the result of an intervention just by its acronym.
– In line 717, “al” is missing after “Han et.”.
Author Response
Reviewer 1
I have read attentively authors responses and amendments and would like to congratulate them on their work. This version has reached high quality on the reporting and discussion/decision-making process of the two cases.
A couple of minor amendments as final recommendation:
– You have not defined minimal clinical important difference upon first appearance in the text. Many readers may not recognize this clinically relevant statistical measure to evaluate the result of an intervention just by its acronym.
– In line 717, “al” is missing after “Han et.”.
Response: Thank you so much for your prompt review and the guidance to sharpen our manuscript. We greatly appreciate it.
We have amended lines 146 to 149 to read as follows:
"Given a minimal clinically important difference (MCID), which is the smallest change in a treatment outcome that an individual patient would identify as important [26], for the Neck Disability Index (NDI) in those without radicular upper extremity pain of 5.5 points, this would be approximately 2.5 MCID levels of change, or an 80% improvement [27]."
We have also added the “al” after the “Han et” in line 461.